# Design and Test of a Sliding Cutting Device for the Plastic Mulch Waste

Mengyu Guo [1,2,3,*], Bin Hu [1,2,3,*], Xin Luo [1,2,3], Chenglin Yuan [1,2,3], Yiquan Cai [1,2,3] and Luochuan Xu [1,2,3]

[1] Xinjiang Production and Construction Corps, Key Laboratory of Modern Agricultural Machinery, Shihezi 832003, China

[2] College of Mechanical and Electrical Engineering, Shihezi University, Shihezi 832000, China

[3] Key Laboratory of Northwest Agricultural Equipment, Ministry of Agriculture and Rural Affairs, Shihezi 832000, China

* Correspondence: shzdxjdxygmy@shzu.edu.cn (M.G.); hb_mac@sina.com (B.H.)

**Abstract:** Agricultural mulch waste that is mechanically recycled has a high resource value. It has been found that the mulch is tightly entangled in the crop straw, forming a knotted feature that prevents further resource utilization. Traditional cutting tools were found to be ineffective in breaking up the knotted feature. In response to the above problems, a sliding cutting device for mechanically recovered mulch waste was proposed and built. The structure of the device and key components were designed and analyzed. A three-factor five-level orthogonal test was conducted and regression variance analysis was performed with the Central Composite Design (CCD) module in Design expert 8. The relationship model was constructed between the test factors such as supporting motor speed $a$, cutting-support rotation speed ratio $b$, and cutting edge angle $c$ and the response indicators such as film breakage rate $y_1$ and knotted feature removal rate $y_2$. The influence law between each key parameter with its significant interaction and the waste crushing effect was analyzed, and the optimum combination of parameters of the crushing device were obtained. Under the same conditions, the errors between the physical test values and the model prediction values of the two response indicators were 2.17% and 3.52%, respectively, indicating that the verification test results were basically consistent with the model prediction results.

**Keywords:** the mulch waste recovered mechanically; crushing of waste; the sliding cutter; knotted feature removal; film breakage rate





## 1. Introduction

Plastic film mulching technology has played an important role in protecting crops and increasing yields in arid, semi-arid, and cold mountainous regions, significantly improving agricultural yields and crop quality [1–5]. However, with the dramatic increase in the use of non-degradable plastic film, a significant amount of mulch waste is left on farmland, which not only pollutes the soil but also causes subtle damage to human health [6–11]. Therefore, there is an urgent need to solve the problem of plastic film pollution on farmland. Mechanized collection is the main technology used to solve the problem of mulch waste pollution on farmland [12–15]. However, the mulch waste collected by mechanization contains a lot of crop straw, and the plastic film is tightly wound on the straw, forming a mixture with the knotted feature shown in Figure 1, which leads to increased cleaning costs and reduced recycling. For example, the Xinjiang region in China uses up to 250,000 tons of agricultural plastic film each year, which could have a direct economic value of over USD 100 million if it were fully recycled and granulated. However, due to its knotted features, most of the film is landfilled and burned [16,17].

To solve the problem of crushing and separation of the plastic film waste, Feng built a film straw shearing and crushing device with a V-shaped cutter to gather, cut, and crush the mixture. The device is more effective in crushing mixtures with more straw content, but the

power consumption is extremely high during the crushing process and it is easy to produce congestion [18]. Haomeng proposed an impeller to disturb the water medium using a film and straw stratification method, and they studied the sinking stratification theory. However, the method is better for crushed film miscellaneous separation, and no relevant equipment was built for further test verification [19]. The above-mentioned scholars have partially explored the method of separating membrane debris mixes, but there is still a lack of corresponding theoretical research and equipment regarding the breaking of the knotted features of membrane and debris.

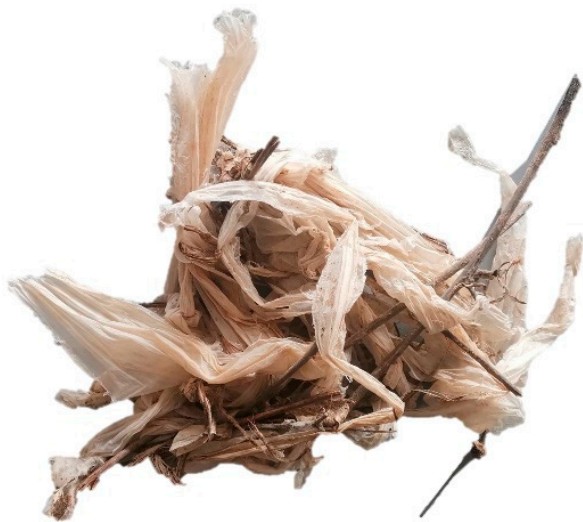

**Figure 1.** The knotted feature.

Residual film and straw are the main components of mechanically recovered waste and have a wide range of applications. Film is the main component that can be recycled for granulation [20,21]; straw is the second main component that can be used to adjust the carbon–nitrogen ratio of compost and to produce animal feed [22,23]; the waste mixture of the two can be used to produce composite panels for the construction and decoration industries [24–26]. However, all of the above resource recovery methods require that the mixed material is first shredded, particularly the knotted feature, to ensure that the subsequent process can be carried out smoothly.

The straw in the mechanized recovery waste has a lower moisture content and a higher degree of lignification than the growing season [27], so it has high stiffness and low toughness, while the residual film is a light and flexible polyethylene material [28], which is more flexible than straw. Therefore, the straw can be effectively crushed by traditional crushing such as impact and shearing, but the residual film will be greatly deformed and difficult to crush due to its own physical properties, and it will be wrapped around the crushing equipment. Not only can its knotted feature not be effectively broken, but the crushing equipment will also be easily damaged.

To solve the above problems, in view of the knotted feature of mulch waste and the difference in physical properties between the residual mulch and straw, a sliding cutter for the mulch waste was proposed and built. An inner arc cutter was used to crush the waste, and the arc-shaped edge can force the waste to slip and deform during the cutting process to remove the knotted feature, which can crush the residual film and straw mixture into a short and uniform broken film and broken straw. The implement used to crush the mulch waste recovered mechanically provides a basis for subsequent resource utilization.

## 2. Materials and Methods

### 2.1. Sliding Cutting Device

2.1.1. The Whole Structure of the Sliding Cutting Device

For the difficult crushing problem of the mulch waste recovered mechanically, a sliding cutting device is built, as shown in Figure 2. The front end cover (9), the rear end cover (6), the cross frame (8), and the support rod (7) constitute the slewing support component; the front end cover (9), the rear end cover (6), the cross frame (8), and the inner arc cutter (11) constitute the slewing sliding cutting part. There is a staggered distribution between the support rod and the sliding cutter without movement interference, and the slewing support part and the slewing sliding cutting part together constitute the crushing part for the crushing function.

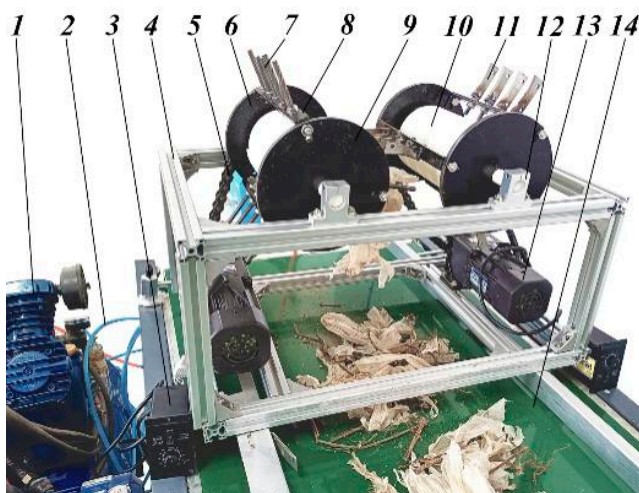

**Figure 2.** Structure of the sliding cutting device, consisting of the following parts: air pump (1), air pipe (2), inverter (3), frame (4), chain (5), rear end cover (6), support rod (7), cross frame (8), front end cover (9), air cavity (10), inner arc cutter (11), vertical optical axis fixed support (12), speed regulating motor (13), conveyor belt (14).

The slewing support part and the slewing cutting part are installed inside with an air cavity (10) that is fixed on the frame (4) through the vertical optical axis fixing support (14) on both sides. In addition, its outlet is always downward and evenly distributed, and it corresponds to the outer support rod (7) and the inner arc cutter (11) one by one. The air pump (1) inputs the high-pressure airflow from the inlet of the air cavity (10) and outputs the airflow from the lower outlet. At the same time, it blows off the wrapping film on the support rod (7) and the sliding cutter (11) passing through the zone to realize the anti-winding function.

2.1.2. The Transmission Part

As shown in Figure 3, the transmission system is designed as follows: the motor (12) drives the driving sprocket (1) to rotate; the driving sprocket (1) drives the driven sprocket (5) to rotate by the chain (2); the driven sprocket (5) and the rear end cover (7) are rigidly connected by the connecting bolt (6); the low-speed thin-walled bearings (3) are installed on the shoulders of the air cavity (9) to support the rotation of the rear end cover (7) and the front end cover (8); at the same time, both sides of the air cavity (9) are fixed and clamped on the frame by the vertical optical axis fixing supports (4). Through the above structural design, the air cavity (9) is fixed, the support part and the sliding cutting part rotate toward each other, and the rotational speed of the transmission part is adjusted by the frequency converter.

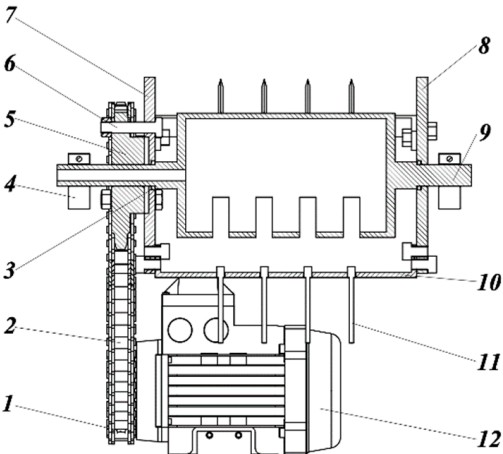

**Figure 3.** Structure of transmission parts, consisting of the following parts: drive sprocket (1), chain (2), low-speed thin-walled bearing (3), vertical optical axis fixed support (4), driven sprocket (5), connecting bolt (6), rear end cap (7), front end cap (8), air cavity (9), cross frame (10), inner arc cutter (11), motor (12).

### 2.1.3. Working Principle

As shown in Figure 4, when the waste is fed in from the top of the device, the left support rod causes it to rotate. When the waste rotates into the effective cutting zone ABA1B1, the inner arc cutter forces it to slide and deform along the arc-shaped edge under the action of the supporting force on both sides. When the material slips to the middle of the arc, the cutting is complete. After cutting, the broken flexible mulch film wound on the support rod or the cutting tool will rotate to the pneumatic anti-tangle zone under the action of the rotary motion, and the wound mulch film will be blown off by the air blowing.

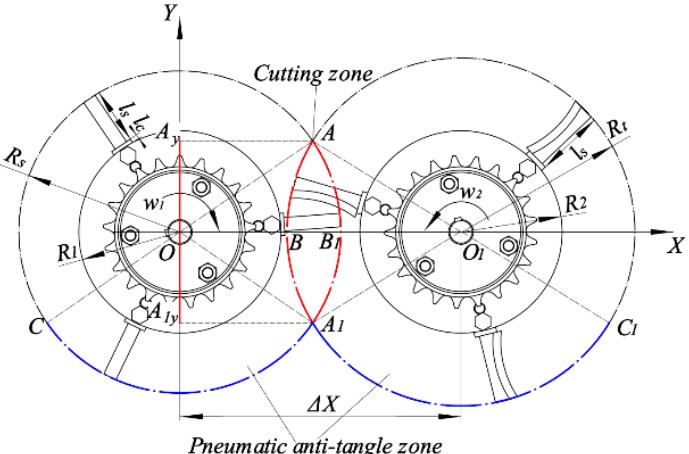

**Figure 4.** Schematic diagram of the working principle zone.

### 2.2. Research on Cutting Part

#### 2.2.1. Structural Design of the Inner Arc Cutter

The cutter is the core part of the device for the crushing of the waste. The crushing object is the waste of straw with high stiffness and low toughness and the film with high flexibility and strong ductility, so the cutter edge of the cutter is designed to be inner-arc-shaped, as shown in the CDEF in Figure 5a. The length of the cutter $l_t = 60$ mm, the width of the cutter $h_t = 20$ mm, the thickness of the cutter $b_t = 4$ mm, and the cutting angle of the edge side $\beta_t = 30°$.

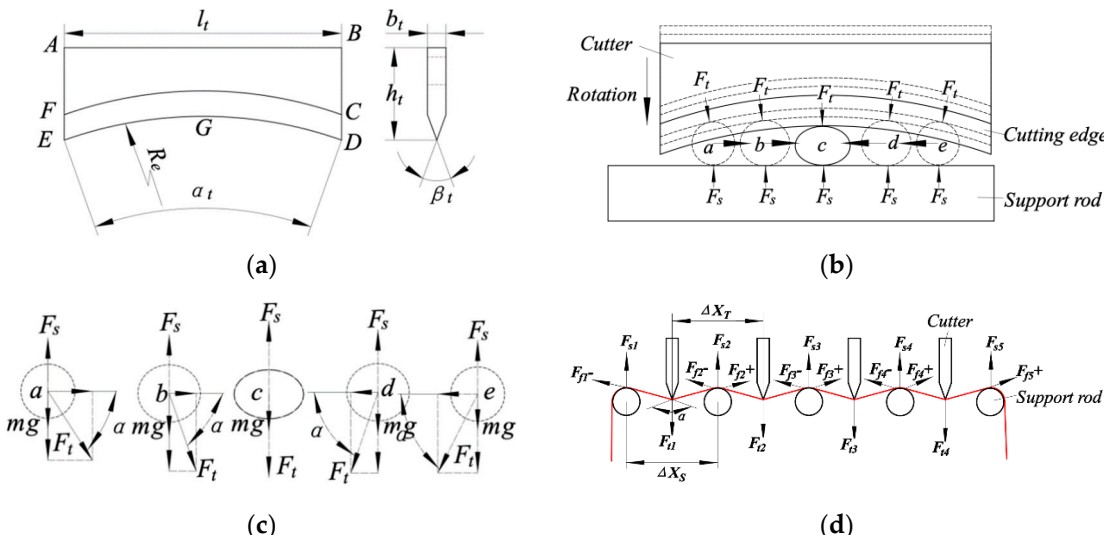

**Figure 5.** Design of inner arc cutter and material displacement and mechanical analysis: (**a**) structure and size parameters of sliding cutter, (**b**) material deformation and displacement, (**c**) material positive force analysis, (**d**) material lateral force analysis.

The structural characteristics of the inner arc cutter that is narrow in the middle and wide on both sides cause the waste to exhibit a sliding agglomeration behavior toward the middle. The sliding behavior has a positive effect on the shearing of flexible film, as shown in Figure 5b. Under the joint action of the cutting force Ft and the supporting force Fs, the agglomeration process will squeeze the waste to deform it, while the cutting force Ft constantly changes its direction of action, forcing the material to move toward the middle.

As shown in Figure 5c, material a and material e on both sides of the arc cutter move toward the position of material b and material d by the component force of the shear force Ft in the horizontal direction. The transient mechanics are analyzed as follows:

$$\begin{cases} F_S = mg + \sum\limits_{i=1}^{n} F_{ti} \cdot sin\alpha \\ \iint \frac{F_t \cdot cos\alpha}{2m} d^2 t = \Delta x \end{cases} \tag{1}$$

where $F_s$ and $F_t$ are the supporting force and the cutting force, respectively; $m$ is the material quality; $\alpha$ is the angle between the shear force and the horizontal direction; $\Delta x$ is material slip displacement.

Materials undergo instantaneous rotary motion and the shear force component undergoes synchronous action, cutting, and extrusion by the arc cutter on the material to force it to produce from the outside to the inside of the aggregation of displacement behavior. To reach the arc cutter mid-point c position, the material force is as follows:

$$\begin{cases} F_S = mg + \sum\limits_{i=1}^{n} F_{ti} \cdot sin\alpha \\ \sum\limits_{i=1}^{2/n} F_{2i-1} \cdot cos\alpha = \sum\limits_{i=1}^{2/n} F_{2i} \cdot cos\alpha \end{cases} \tag{2}$$

According to the formula shown above, the angle $\alpha$ between the shear force and the horizontal direction will have an impact on the material slip behavior, and the angle $\alpha$ is influenced by the cutter edge angle $\alpha t$ in the cutting process. Therefore, the smaller the $\alpha t$, the smaller the angle $\alpha$, which leads to a more likely slip behavior, but the decrease in the effective shear force, that is, the vertical component of the shear force $F_t$, lowers the crushing effect on rigid materials such as straw. The larger the $\alpha_t$, the smaller the $\alpha$, the smaller the component force of $F_t$ in the horizontal direction, the greater the component

force in its vertical direction, and the greater the effective shear force, but it is difficult to exhibit slip behavior, and the crushing effect on the ground film is poor. To improve the cutting effect of the waste, the specific parameters of the cutter edge angle $\alpha_t$ need to be further analyzed. According to the above analysis results, the value range of $\alpha_t$ is set between 22.5° and 67.5°.

### 2.2.2. Structural Analysis of Support Rods

Figure 5d shows the lateral mechanical analysis of the crushing, and $F_s$, $F_t$, and $F_f$ are the supporting force, the cutting force, and the friction force in the material crushing process, respectively. From the picture, we can find that, in the cutting process, the contact mode of the cutter with the material is line contact, and the contact mode of the support rod with the material is surface contact, so it can achieve the effect of effective support. The formula is as follows:

$$
\begin{cases}
\sum\limits_{5}^{i=1} F_{si} = \sum\limits_{4}^{j=1} F_{tj} \\
F_{si} = F_{fi-} \cdot \cos\frac{\alpha}{2} + F_{fi+} \cdot \cos\frac{\alpha}{2} \, (1 < i < 5) \\
F_{s1} = F_{f1-} \cdot \cos\frac{\alpha}{2} \\
F_{s5} = F_{f5+} \cdot \cos\frac{\alpha}{2}
\end{cases}
\tag{3}
$$

where $F_s$, $F_t$, and $F_f$ are the supporting force, cutting force, and friction force in the process of material crushing, N; $\alpha$ is the deformation angle of the material under the action of the cutter, °; $i$ and $j$ are the force's label from left to right, respectively.

As the distance of the cutter over the support rod under the action of the rotary force is greater, the angle $\alpha$ is smaller, and the cutting force of the bidirectional support on the material is greater. When the cutting force is greater than the tensile strength of the material itself, the local position where the material contacts the cutter is broken, and the material is crushed. The analysis shows that the support force on the material has an important relationship with the friction force between the material and the support rod. Therefore, choosing a support rod with a larger friction force has a positive effect on the cutting of the material. Therefore, the threaded rod with an outer diameter of 8 mm × 70 mm is selected as the support rod. The thread groove on the surface of the threaded rod has a positive effect on increasing the friction force of the material, and the structure of the threaded rod will facilitate installation on the cross bar.

### 2.2.3. Energy Consumption Analysis with Different Cutting Conditions

The cutting parts of the crushing bench of the film and impurity waste recovered mechanically consist of the support part and the cutting part, as shown in Figure 6, which can be divided into two cutting conditions according to their rotating speed difference: passive slow-speed cutting with the slide cutter installed in reverse and active fast-speed cutting with the slide cutter installed in forward.

$$
\begin{cases}
J_1 = m_1 r^2 \\
J_2 = \int_{R_1+l_c}^{R_s} \frac{m_2}{l_s} r^2 dl \\
J_3 = \int_{R_2+l_c}^{R_t} \frac{m_3}{l_t} r^2 dl \\
J_4 = \int_{R_0}^{R_1} r^2 \sigma 2\pi r dr \\
J_S = aJ_1 + bJ_2 + cJ_4 \\
J_T = aJ_1 + dJ_3 + cJ_4 \\
P = \frac{J_s\omega_1 + J_T\omega_2}{2t}
\end{cases}
\tag{4}
$$

where $J_1$, $J_2$, $J_3$, $J_4$, $J_S$, and $J_T$ are the rotational inertia of the cross frame, support bar, cutter, end cap, support part, and cutting part, respectively; $r$, $R_s$, $R_t$, $R_1$, and $R_0$ are the radius

of the center of the cross frame, the radius of the support bar, the radius of the cutter, the radius of the end cap, and the radius of the inner hole of the end cap, respectively; $m_1$, $m_2$, $m_3$, and $m_4$ are the masses; $a$, $b$, $c$, and $d$ are the number of cross frames, support bars, cutting knives, and end caps, respectively; $\omega_1$ and $\omega_2$ are the angular velocity of the support part and cutting part, respectively; $t$ is the system running time; $P$ is the total power of the system.

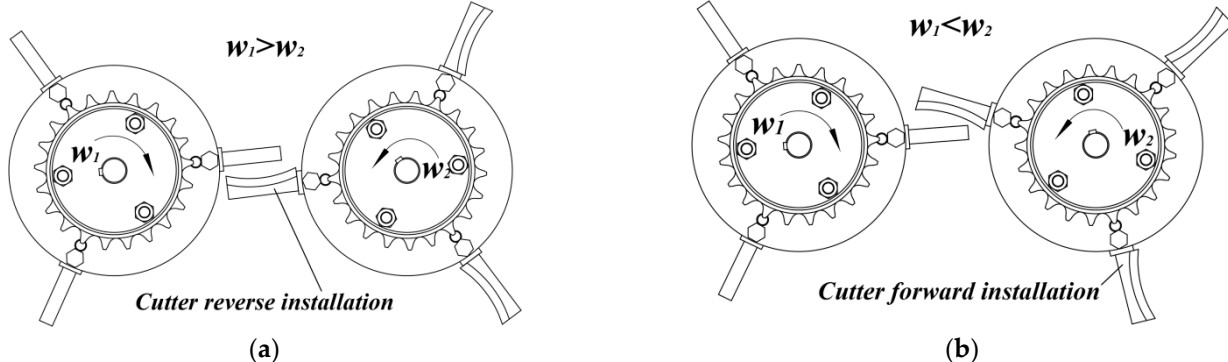

**Figure 6.** Different installation conditions of the cutter: (**a**) passive slow-speed cutting with the slide cutter installed in reverse, (**b**) active fast-speed cutting with the slide cutter installed in forward.

The difference in system power between the two types of cutting is as follows:

$$\Delta P = P_S - P_T = \frac{J_s\omega_1 + J_t\omega_2 - J_s\omega_2 - J_t\omega_1}{2t} = \frac{(bJ_2 - dJ_3)\omega_1 + (dJ_3 - bJ_2)\omega_2}{2t} \quad (5)$$

The support rod and the cutter use 304 stainless steel as the base material, with support rod mass $m_2 = \pi r^2 l_s \rho$ and cutter mass $m_3 < l_1 l_2 l_3 \rho$, and in the numerical calculation, $m_2 > l_1 l_2 l_3 \rho > m_3$ due to $R_1 = R_2$, $J_2 > J_3$ due to $b = d + 1$, and $\omega_1 > 3\omega_2$. Therefore, the power consumption of passive slow-speed cutting with the slide cutter installed in reverse is larger than active fast-speed cutting with the slide cutter installed in forward. This paper chooses the active fast cutting with the slide cutter mounted forward.

2.2.4. Analysis of Cutting and Supporting Rotation Speed Ratio with Effective Cutting

The cutting zone of the waste in the crushing device is the red area marked as $ABA_1B_1$ in Figure 4. When the support rod and the cutter pass through this zone and complete the cutting behavior, the coordinates of the top of the support rod and the top of the cutter in the Y-axis projection between $A_y$ and $A_y'$ will coincide.

If the support rod passes through this zone and the cutter in the Y-axis projection does not coincide with it, it means that the material on this support rod is missing cutting and there is invalid cutting behavior. The cutting zone equation is as follows:

$$\begin{cases} x^2 + y^2 = R_s^2 \\ (x - \Delta X)^2 + y^2 = R_t^2 \\ R_s = R_1 + l_c + l_s \\ R_t = R_2 + l_c + l_t \end{cases} \quad (6)$$

where $R_s$, $R_t$, $R_1$, and $R_2$ are the support bar tip outer diameter, cutter tip outer diameter, support bar end cap radius, and cutter end cap radius (mm), respectively; $l_c$, $l_s$, and $l_t$ are the cross frame and end cap minimum distance, support bar length, and cutter length (mm), respectively; $\Delta X$ is the center distance, mm.

The solution is obtained by substituting the values into the equation: $x = 118.15$, $y = \pm 80.79$; $-80 < y < 80$ is selected as the boundary condition, and the Solidworks-Motion module is used to simulate the cutting process of the crushing device within the boundary condition, with the top point of the support rod and the cutter tip as the data collection

point. The rotation speed of the support rod is set to 60 rpm, the projection of the top of the support rod and the tip of the cutter on the Y-axis is obtained by adjusting the rotation speed of the cutter, and the unstable phase data are removed during the acceleration of the motor turning on, according to the boundary conditions, to obtain the minimum cutting-support rotation speed ratio (C-S speed ratio) of effective cutting.

The Y-axis projection trajectory of the simulation results is shown in Figure 7. The results show that: (1) when the C-S speed ratio is 1.2:1, there are 2 missed cuts in one rotation cycle; (2) when the C-S speed ratio is 1.5:1, there are 2 missed cuts in one rotation cycle; (3) when the C-S speed ratio is 2:1, there is 1 missed cut in one rotation cycle; (4) when the C-S speed ratio is 3:1, the support rod does not exhibit the phenomenon of missing cuts in one rotation cycle, and as the speed increases to 4:1 and 5:1, not only are there no missing cuts, but repeating cuts also appear at 5:1. Therefore, the simulation analysis shows that the minimum of C-S speed ratio to achieve effective cutting is 3:1.

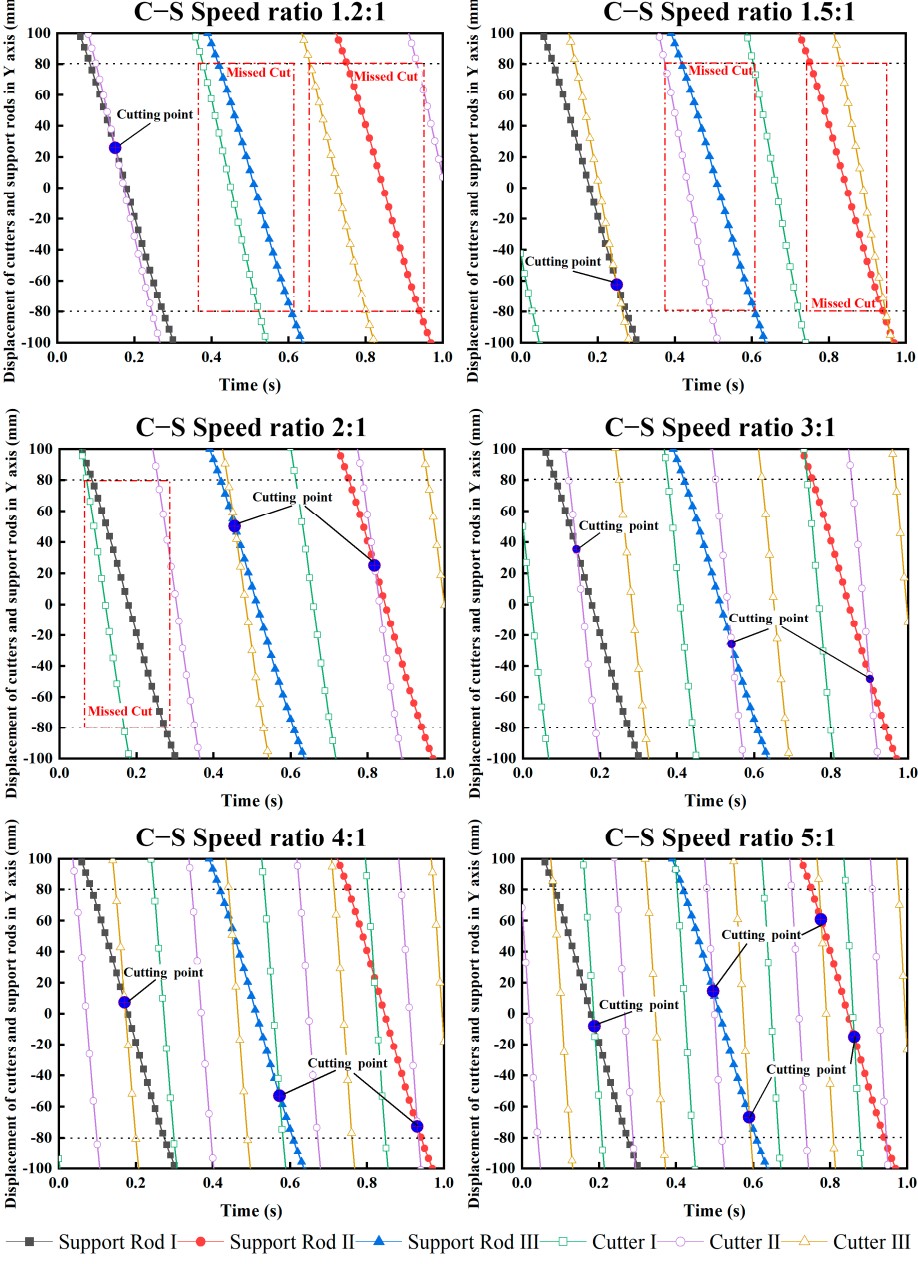

**Figure 7.** Simulation results of different speed ratios.

### 2.3. Test Devices, Methods, and Response Indicators

After construction of the crushing bench of internal blowing and sliding cutting for mulch waste recovered mechanically, experiments were carried out in the Precision Agriculture Technology and Equipment Laboratory of Beiyuan New Area, Shihezi University.

The test material was selected from the experimental field of 4th Company, 136th Regiment, Karamay District, Karamay City, Xinjiang Uygur Autonomous Region, China (longitude: 84.86°, latitude: 45.59°), where cotton was planted in one film and six rows, with a film thickness of 0.01 mm and mulch width of 2050 mm. The cotton had already been harvested mechanically and the drip irrigation tape under the film had been extracted.

Three rolls of the mulch wastes were randomly selected and collected after the operation by a spring-teeth film recycling machine. The main instruments and equipment used during the test included: a JA4100 electronic balance (measuring range: 0~1000 g, measuring accuracy: 0.001g, supplier: Shanghai Sunyu Hengping Instrument Co., Ltd., Shanghai, China), TCS-300 electronic bench scale (measuring range: 0~300 kg, measuring accuracy: 2 g, supplier: Shanghai Caixin Electronics Co., Shanghai, China), bags, and a self-built test crushing device of internal blowing and sliding shearing for mulch waste.

When the material test was carried out, the film waste was first weighed in advance to $1 \pm 0.05$ kg, the crushing test bench was turned on, and the waste was put into the bench directly above after the equipment was run smoothly.

After crushing, the crushed material at the outlet of the device was manually sampled and sorted, the completed crushed film was recovered and weighed as $m_1$, and the waste with an unbroken knotted feature was removed and weighed as $m_2$.

Response Indicator 1 film breakage rate $y_1$ for the mechanized recovered film and impurity waste is as follows:

$$y_1 = \frac{m_1}{m} \times 100\% \tag{7}$$

Response Indicator 2 knotted feature removal rate $y_2$ for the mechanized recovered waste is as follows:

$$y_2 = \frac{m - m_2}{m} \times 100\% \tag{8}$$

where $m$, $m_1$, and $m_2$ are the total mass of the sample, the mass of the broken film, and the mass of the knotted feature, respectively, in g; $y_1$ and $y_2$ are the film breakage rate and the knotted feature removal rate of the waste, respectively, in %.

### 2.4. Test Scheme

According to previous research, the value of the support motor speed $n_1$(a) was 60~180 r·min$^{-1}$, the C-S speed ratio $\lambda$(b) was 3~7, and the cutter edge angle $\alpha_t$(c) was 22.5°~67.5°. The crushing effect of the membrane impurity waste was better and the above test factors were the key factors, and a three-factor five-level orthogonal test was carried out with film crushing rate $y_1$ and knotted feature removal rate $y_2$ as the response indicators. Using the Central Composite Design (CCD) module in the software Design expert 8, a three-factor five-level orthogonal test table was designed as shown in Table 1: The test consisted of 14 groups of analysis factor tests and 5 groups of center zero estimation error tests, a total of 19 groups.

**Table 1.** Three-factor five-level central combination test scheme and results.

| S/N | $a$ (r·min$^{-1}$) | $b$ | $c$ (°) | $y_1$ (%) | $y_2$ (%) |
|-----|-----|-----|-----|-----|-----|
| 1 | −1 (60) | −1 (3) | −1 (22.5) | 76.8 | 74.1 |
| 2 | 1 (180) | −1 | −1 | 78.3 | 75.3 |
| 3 | −1 | 1 (7) | −1 | 70.3 | 80.0 |
| 4 | 1 | 1 | −1 | 72.4 | 78.9 |
| 5 | −1 | −1 | 1 (67.5) | 72.1 | 77.1 |
| 6 | 1 | −1 | 1 | 73.1 | 75.3 |
| 7 | −1 | 1 | 1 | 84.6 | 79.9 |

**Table 1.** *Cont.*

| S/N | $a$ (r·min$^{-1}$) | $b$ | $c$ (°) | $y_1$ (%) | $y_2$ (%) |
|-----|------|------|------|------|------|
| 8 | 1 | 1 | 1 | 86.7 | 74.3 |
| 9 | −1.682 (19.09) | 0 (5) | 0 | 87.3 | 81.1 |
| 10 | 1.682 (220.91) | 0 | 0 | 89.7 | 78.3 |
| 11 | 0 (120) | −1.682 (1.64) | 0 (45) | 68.3 | 65.1 |
| 12 | 0 | 1.682 (8.36) | 0 | 73.1 | 67.5 |
| 13 | 0 | 0 | −1.682 (7.16) | 81.2 | 88.3 |
| 14 | 0 | 0 | 1.682 (82.84) | 84.3 | 89.9 |
| 15 | 0 | 0 | 0 | 89.3 | 90.3 |
| 16 | 0 | 0 | 0 | 90.4 | 91.2 |
| 17 | 0 | 0 | 0 | 88.2 | 89.3 |
| 18 | 0 | 0 | 0 | 89.1 | 92.6 |
| 19 | 0 | 0 | 0 | 91.2 | 90.5 |

Note: $a$, $b$, and $c$ represent, respectively, the support motor speed, the C-S speed ratio, and the cutter edge angle; $y_1$ and $y_2$ represent, respectively, the film breakage rate and the knotted feature removal rate.

## 3. Results

### 3.1. Regression Analysis of Variance and Model Construction

Regression analysis of variance (ANVOA) on the test results of Table 1 was carried out by the Analysis module of the software Design expert 8.

The results from the regression analysis of variance are shown in Table 2. According to the results of the regression analysis of variance of the test response indicator y1 in Table 2, it can be seen that the $p$ values of the interactive factors $bc$, $b^2$, and $c^2$ were all below 0.01, which were extremely significant influencing factors for $y_1$. The $p$ value of the single factors $b$ and $c$ ranged from 0.01 to 0.05, which were significant influencing factors for $y_1$. The $p$ values of other factors were all above 0.05, which were non-significant influencing factors for $y_1$.

**Table 2.** Regression analysis of variance of the test results.

| Source | $y_1$ | | $y_2$ | |
|-----|------|------|------|------|
| | $F$-Value | $p$-Value | $F$-Value | $p$-Value |
| Model | 22.65 | <0.0001 ** | 108.79 | <0.0001 ** |
| $a$ | 1.60 | 0.2383 | 8.43 | 0.0175 * |
| $b$ | 6.56 | 0.0306 * | 13.74 | 0.0049 ** |
| $c$ | 7.91 | 0.0203 * | 0.057 | 0.8161 |
| $ab$ | 0.068 | 0.7997 | 3.71 | 0.0862 |
| $ac$ | $5.907 \times 10^{-3}$ | 0.9404 | 5.61 | 0.0420 * |
| $bc$ | 35.02 | 0.0002 ** | 5.91 | 0.0379 * |
| $a^2$ | 3.18 | 0.1082 | 181.08 | <0.0001 ** |
| $b^2$ | 141.42 | <0.0001 ** | 846.30 | <0.0001 ** |
| $c^2$ | 25.49 | 0.0007 ** | 6.19 | 0.0345 * |
| Lack of Fit | 6.14 | 0.0516 | 0.71 | 0.6492 |
| | $R^2 = 0.9577$ | | $R^2 = 0.9909$ | |
| | $R^2_{adj} = 0.9154$ | | $R^2_{adj} = 0.9818$ | |
| | C.V. = 2.83% | | C.V. = 1.38% | |

Note: * indicates significant ($0.01 < p < 0.05$), ** indicates extremely significant ($p < 0.01$). The full form of C.V. is the coefficient of variation.

Ignoring the non-significant influencing factors of the test response indicator $y_1$, according to the $p$ value significance, the extremely significant and significant factors of $y_1$ were ranked as follows: $b^2 > bc > c^2 > c > b$.

From the results of the regression analysis of variance of the test response indicator $y_2$ in Table 2, it can be seen that the $p$ values of the single factor $b$ and the interactive factors $a^2$ and $b^2$ were all below 0.01, which were extremely significant factors for $y_2$. The $p$ value

of the single factor $a$ and the interactive factors $ac$, $bc$, and $c^2$ all ranged from 0.01 to 0.05, which were significant factors for $y_1$. The $p$ values of other factors were all above 0.05, which were non-significant influencing factors for $y_2$.

Ignoring the non-significant influencing factors of the test response indicator $y_2$, according to the significance of the $p$ value, the extremely significant and significant factors of $y_2$ were ranked as follows: $a^2 = b^2 > b > a > bc > ac$.

According to the results of regression analysis of variance, the constructed second-order response models $Y_1$ and $Y_2$ between the test influencing factors $a$, $b$, and $c$ and the test response indicators $y_1$ and $y_2$ were:

$$
\begin{cases}
Y_1 = 89.77 + 0.79a + 1.59b + 1.75c + 0.21ab - \\
\quad 0.06ac + 4.81bc - 1.11a^2 - 7.40b^2 - 3.14c^2 \\
Y_2 = 90.81 - 0.88a + 1.12b + 0.073c - 0.76ab - \\
\quad 0.94ac - 0.96bc - 4.08a^2 - 8.82b^2 - 0.75c^2
\end{cases}
\tag{9}
$$

The $p$ values of the model coefficients of $Y_1$ and $Y_2$ were both below 0.01, indicating that the constructed second-order response models $Y_1$ and $Y_2$ were extremely significant. In addition, the coefficient of determination $R^2$, corrected coefficient of determination $R^2_{adj}$, and coefficient of variation C.V. of model $Y_1$ were 0.9577, 0.9154, and 2.83%, respectively, indicating that the constructed model $Y_1$ had a high degree of interpretation, and the model could be used to predict the value of $y_1$ accurately and reliably. The coefficient of determination $R^2$, corrected coefficient of determination $R^2_{adj}$, and coefficient of variation C.V. of model $Y_2$ were 0.9909, 0.9818 and 1.38%, respectively, indicating that the constructed model $Y_2$ had a high degree of interpretation, and the model could be used to predict the value of $y_2$ accurately and reliably.

### 3.2. Analysis of the Influence Law of Single Factor on the Crushing of Waste

To observe the influence of the test factors $a$, $b$, and $c$ on the crushing of the mulch waste recovered mechanically, the Analysis module in Design expert was used to obtain the influence law of a single factor on the film breakage rate and knotted feature removal rate in waste. The single-factor model curve redrawn using Origin data processing software is shown in Figure 8.

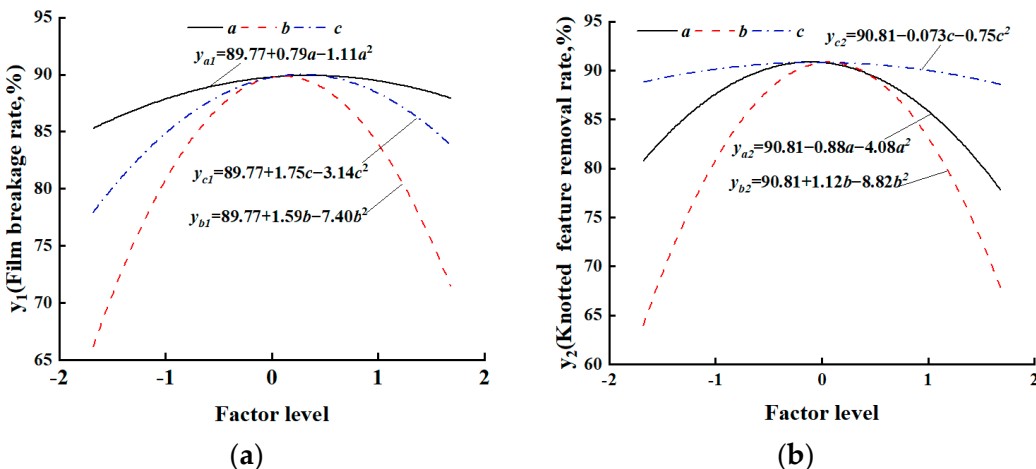

**Figure 8.** Influence law of single factors on crushing effect of waste: (**a**) Influence law of single factors on $y_1$; (**b**) Influence law of single factors on $y_2$.

#### 3.2.1. Influence Law of Single Factors $a$, $b$, and $c$ on Film Breakage Rate $y_1$

In Figure 8a, $y_{a1}$, $y_{b1}$, and $y_{c1}$ are the fitting curves of the influence of test factors $a$, $b$, and $c$ on the response indicator $y_1$, respectively. When the levels of test factors $a$, $b$, and $c$ were between 0 and 1, the film breakage rate $y_1$ was at its peak value, where the level of

factor a was close to 0, the level of factor *b* was close to 0.3, and the level of factor *c* was close to 0.3. When approaching 0.5, the three curves approached the peak of $y_1$. When the level values of factors *a*, *b*, and *c* approached −1.682 and 1.682, $y_1$ was in a lower position in the three curves. At this time, the curve was $y_{a1} > y_{c1} > y_{b1}$ and the steepness of the overall curve was $y_{b1} > y_{c1} > y_{a1}$, so the ranking of the influence of a single factor on the film breakage rate was $b > c > a$.

### 3.2.2. Influence Law of Single Factors *a*, *b*, and *c* on Knotted Feature Removal Rate $y_2$

In Figure 8b, $y_{a2}$, $y_{b2}$, and $y_{c2}$ are the fitting curves of the influence of test factors *a*, *b*, and *c* on the response indicator $y_2$, respectively. When the levels of test factors *a*, *b*, and *c* were between −0.5 and 0.5, the knotted features removal rate $y_2$ was at its peak value, where the level of factor *a* was close to 0, the level of factor *b* was close to 0.2, and the level of factor *c* was close to 0. When the level of *c* approached 0, the three curves approached the peak value of $y_2$. When the level values of the factors *a*, *b*, and c approached −1.682 and 1.682, $y_2$ was in a lower position in the three curves. At this time, the curve was $y_{c1} > y_{a1} > y_{b1}$ and the steepness of the overall curve was $y_{b1} > y_{a1} > y_{c1}$, so the ranking of the influence of a single factor on the film breakage rate was $b > a > c$.

### 3.3. Influence Law of Significant Interactive Factor on the Crushing of Waste

It can be seen from the regression analysis of variance in Table 2 that the interactive factor *bc* was an extremely significant factor affecting the test response indicator $y_1$. The interactive factors *ac* and *bc* were the significant influencing factors of the test response indicator $y_2$. The non-significant interactive factor was ignored, and only the influences of the extremely significant and significant interactive factor on the values of the test response indicators $y_1$ and $y_2$ were analyzed.

### 3.3.1. Influence Law of Significant Interactive Factor on Response Indicator $y_1$

Figure 9 shows the response surface graph of the significant interactive factor *bc* to the response indicator $y_1$. *b* (C-S rotation speed ratio $\lambda$) was 1.64~8.36, *c* (the cutter edge angle $\alpha_t$) was 7.16~82.84°, and the corresponding factor level was −1.682~1.682.

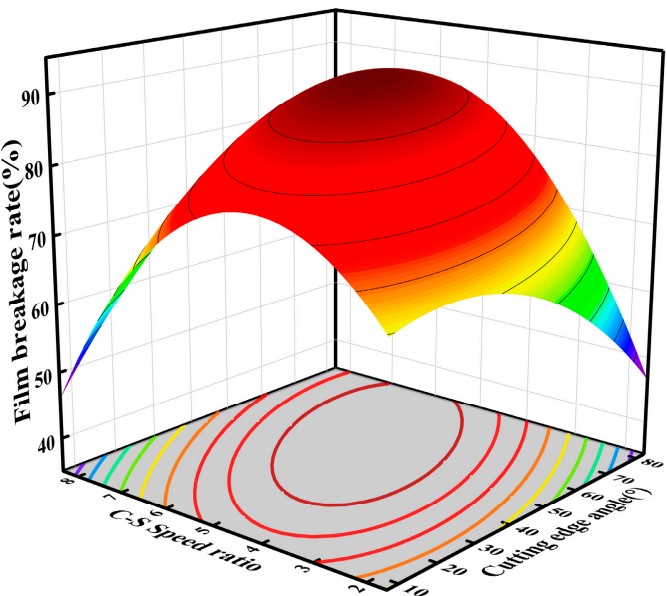

**Figure 9.** Influence law of significant interactive factors on $y_1$.

When the test factor *b* level value was −1.682 and the factor *c* level value increased from −1.682 to 1.682, $y_1$ first increased from 67.95% at 7.16° to 69.38% at 21.81°, and then gradually decreased to 46.65% at 82.84°, showing a trend of first slowly increasing and then

rapidly decreasing, and the decrease was more obvious than the increase. When the test factor $b$ level value was 1.682 and the factor $c$ level value increased from $-1.682$ to 1.682, $y_1$ first increased from 46.12% at 7.16 to 79.20% at 82.84°, showing an increasing trend all the time.

Compared with the influence trend of the single factor $c$ on $y_1$, the variation law of $y_1$ obtained under the above conditions was quite different from the influence variation rule of the single factor $c$ on $y_1$, indicating that the test factor $b$ still had a significant effect on $y_1$ under this condition. From the above analysis, it can be seen that when the value of $b$ (C-S speed ratio $\lambda$) is low, it is better to choose a lower $c$ (cutter edge angle $\alpha_t$) to obtain a higher $y_1$; when the value of $b$ (C-S speed ratio $\lambda$) is high, choosing a higher $c$ (cutter edge angle $\alpha_t$) is good for obtaining a higher $y_1$.

When the level value of the test factor $c$ was $-1.682$ and the level value of the factor $b$ increased from $-1.682$ to 1.682, $y_1$ first increased from 67.95% at 1.64 to 79.33% at 4.02 and then gradually decreased to 46.11% at 8.36, showing a trend of increasing first and then decreasing. The trend of increasing and decreasing was close, but the magnitude of the decrease was larger than that of the increase. When the level value of the test factor $c$ was 1.682, and the level value of the factor $b$ increased from $-1.682$ to 1.682, $y_1$ first increased from 46.64% at 1.64 to 89.97% at 6.41 and then gradually decreased to 79.20% at 8.36, showing a trend of increasing first and then decreasing, but the increasing trend was greater than that of the decreasing trend.

Compared with the influence trend of single factor $b$ on $y_1$, the variation rule of the $y_1$ value obtained under the above conditions was close to the influence variation rule of single factor $b$ on $y_1$. Based on the above analysis, it can be seen that, when the value of $c$ (cutter edge angle $\alpha_t$) is low, it is better to choose a lower $b$ (C-S speed ratio $\lambda$) to obtain a higher value of $y_1$; when the value of $c$ (cutter edge angle $\alpha_t$) is high, selecting a higher $b$ (C-S speed ratio $\lambda$) is good for obtaining a higher $y_1$.

In addition, under the double action of test factors $b$ and $c$, when the level value of the two factors increased from $-1.682$ to 1.682 at the same time, $y_1$ showed a trend of increasing first and then decreasing. It can be seen from the response surface projection contour changes in Figure 9 that when $b$ (C-S speed ratio $\lambda$) was 5.5 and $c$ (cutter edge angle $\alpha_t$) was 55°, a larger $y_1$ could be obtained.

### 3.3.2. Influence Law of Significant Interactive Factor on Response Indicator $y_2$

(1)  Influence law of ac on $y_2$

Figure 10a shows the response surface of the significant interactive factor $ac$ to the response indicator $y_2$, where $a$ (support motor speed $n_1$) was 19.09~220.91 r·min$^{-1}$, $c$ (cutter edge angle $\alpha_t$) was 7.16~82.84°, and the corresponding factor level was $-1.682$~1.682.

When the level value of the test factor $a$ was $-1.682$ and the level value of the factor $c$ increased from $-1.682$ to 1.682, $y_2$ changed slowly from 75.84% at 7.16° to 81.40% at 82.84°, which always showed an increasing trend, but the trend was relatively slow. When the level value of the test factor $a$ was 1.682 and the level value of the factor $c$ increased from $-1.682$ to 1.682, $y_2$ first slowly increased from 78.20% at 7.16° to 78.55% at 21.807° and then slowly decreased to 73.13% at 82.84°. In the process, it showed a trend of first slowly increasing and then slowly decreasing, and the decreasing range was larger than the increasing range.

Compared with the influence trend of the single factor $c$ on $y_2$, the variation rule of $y_2$ obtained under the above conditions was close to the influence variation rule of the single factor $c$ on $y_2$. Based on the above analysis, it can be seen that when the value of $a$ (support motor speed $n_1$) is low, it is better to choose a higher $c$ (cutter edge angle $\alpha t$) to obtain a higher $y_2$; when the value of $a$ (support motor speed $n_1$) is high, selecting a higher value of $c$ (cutter edge angle $\alpha_t$) is good for obtaining a higher $y_2$.

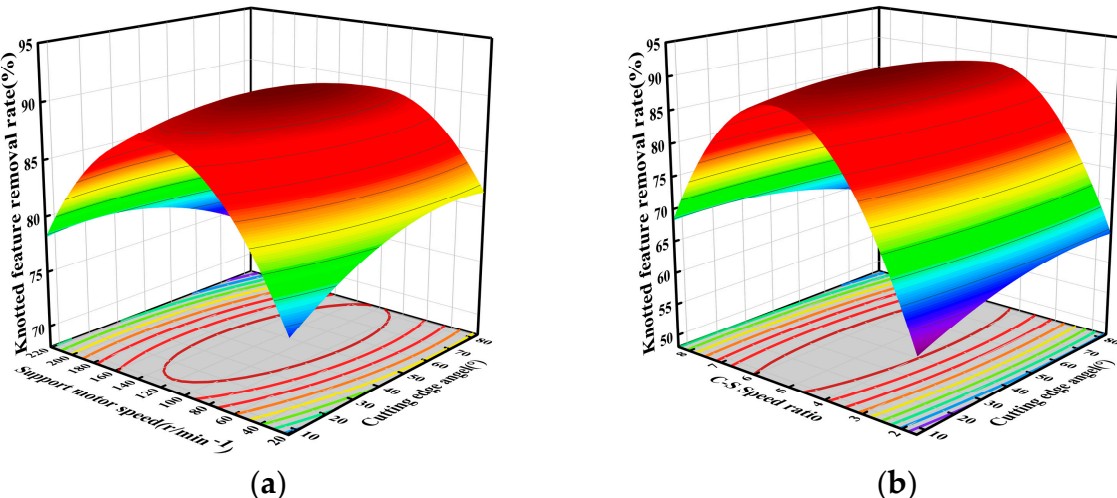

**Figure 10.** Influence law of significant interactive factors on $y_2$: (**a**) Influence law of *ac* on $y_2$, (**b**) Influence law of *bc* on $y_2$.

When the test factor *c* level value was $-1.682$ and the factor a level value increased from $-1.682$ to $1.682$, $y_2$ first increased from 75.84% at 19.09 r·min$^{-1}$ to 88.58% at 123.26 r·min$^{-1}$ and then gradually decreased to 78.19% at 220.91 r·min$^{-1}$, showing the trend of first increasing and decreasing. The trend between increasing and decreasing was analogous, but the increasing range was larger than the decreasing range. When the test factor *c* level value was 1.682 and the factor *a* level value increased from $-1.682$ to $1.682$, $y_2$ first increased from 81.40% at 19.09 r·min$^{-1}$ to 89.16% at 103.72 r·min$^{-1}$ and then gradually decreased to 73.13% at 220.91 r·min$^{-1}$, showing the trend of increasing first and decreasing, but the increasing trend was smaller than the decreasing trend.

Compared with the influence trend of the single factor *a* on $y_2$, the variation law of the $y_2$ value obtained under the above conditions was different from the influence variation law of the single factor *a* on $y_2$, indicating that in the above conditions, the test factor *c* still had a significant effect on $y_2$ From the above analysis, it can be seen that when the value of *c* (cutter edge angle $\alpha_t$) is low, it is better to choose a lower *a* (support motor speed $n_1$) to obtain a higher $y_2$; when the value of *c* (cutter edge angle $\alpha_t$) is high, selecting a higher *a* (support motor speed $n_1$) is good for obtaining a higher $y_2$.

In addition, under the collective effect of test factors *a* and *c*, when the level value of two factors increased from $-1.682$ to $1.682$ at the same time, the $y_2$ value showed a trend of increasing first and then decreasing. It can be seen from the change in the contour line of the response surface projection in Figure 10a that when *a* (support motor speed $n_1$) was 110 r·min$^{-1}$ and *b* (C-S speed ratio $\lambda$) was 51°, a larger $y_2$ value could be obtained.

(2)    Influence law of *bc* on $y_2$

When the test factor *b* level value was $-1.682$ and the factor *c* level value increased from $-1.682$ to $1.682$, $y_2$ first changed from 59.06% at 7.16° to 64.94% at 70.63° and then gradually decreased to 64.69% at 82.84°, showing a trend of first increasing and then decreasing. The increasing trend was more obvious than the decreasing trend. When the test factor *b* level value was 1.682 and the factor *c* level value increased from $-1.682$ to $1.682$, $y_2$ first increased from 68.25% at 7.16° to 68.52% at 21.81° and then gradually decreased to 62.77% at 82.84°, showing a trend of first increasing and then decreasing, but the decreasing trend was more obvious than the increasing trend.

Compared with the influence trend of the single factor *c* on $y_2$, the variation law of the $y_2$ value obtained under the above conditions was consistent with the influence variation law of the single factor *c* on $y_2$. From the above analysis, it can be seen that when the value of *b* (C-S speed ratio $\lambda$) is low, it is better to choose a higher *c* (cutter edge angle $\alpha_t$) to obtain

a higher value of $y_2$; when the value of $b$ (C-S speed ratio $\lambda$) is high, selecting a lower $c$ (cutter edge angle $\alpha_t$) is beneficial to obtain a higher $y_2$.

When the level value of the test factor $c$ was $-1.682$ and the level value of the factor $b$ increased from $-1.682$ to $1.682$, $y_2$ value first increased from 59.06% at 1.64 to 88.75% at 5.33 and then gradually decreased to 68.25% at 8.36, showing a trend of increasing first and then decreasing. When the test factor $c$ level value was $1.682$ and the factor $b$ level value increased from $-1.682$ to $1.682$, $y_2$ first increased from 64.69% at 1.64 to 88.62% at 4.89 and then gradually decreased to 62.77% at 8.36, showing a trend of first increasing and then decreasing.

Compared with the influence trend of the single factor $b$ on $y_2$, the variation law of $y_2$ obtained under the above conditions was close to the influence variation law of the single factor $b$ on $y_2$. From the above analysis, it can be seen that when the value of $c$ (cutter edge angle $\alpha_t$) is low, it is better to choose a lower $b$ (C-S speed ratio $\lambda$) to obtain a higher $y_2$; when the value of $c$ (cutter edge angle $\alpha_t$) is high, selecting a moderate $b$ (C-S speed ratio $\lambda$) is beneficial for obtaining a higher $y_2$.

In addition, under the double action of test factors $b$ and $c$, when the level value of the two factors increased from $-1.682$ to $1.682$ at the same time, the $y_2$ value showed a trend of first increasing and then decreasing. It can be seen from the change in the projection contour line of the response surface in Figure 10b that when $b$ (C-S speed ratio $\lambda$) was 5 and $c$ (cutter edge angle $\alpha_t$) was 43, a larger $y_2$ could be obtained.

### 3.4. Optimization of Target Parameters and Experimental Validation

In the separation process of mulch waste in mechanized recovery, the better the film crushing effect is, the less knot-like the features are, and the more conducive to the separation of membrane impurity waste. To obtain a better crushing effect, the key parameters such as $a$ (support motor speed $n_1$), $b$ (C-S speed ratio $\lambda$), and $c$ (cutter edge angle $\alpha_t$) were optimized to obtain the optimal combination of key parameters.

The built quadratic regression model was optimized and analyzed by the Optimization module in the Design Expert data analysis software, and the constraints were:

$$\begin{cases} \max(y_1, y_2) \\ s.t. \begin{cases} 19.09 \leq a \leq 220.91 \\ 1.64 \leq b \leq 8.36 \\ 7.16 \leq c \leq 82.84 \end{cases} \end{cases} \tag{10}$$

According to the optimization results, the optimal parameter combination was selected as follows: when the support motor speed $n_1$ was 118.03 r·min$^{-1}$, the C-S speed ratio $\lambda$ was 5.25, and the cutter edge angle $\alpha_t$ was 52°, the prediction of the corresponding indicators $y_1$ and $y_2$ was tested. The values were 90.25% and 90.76%.

On 16 December 2021, a verification test was carried out in the Precision Agriculture Technology and Equipment Laboratory of Shihezi University. During the test, the support motor speed $n_1$ was selected as 120 r·min$^{-1}$, the C-S speed ratio $\lambda$ was 5.3, and the cutter edge angle $\alpha_t$.was 52°. The analysis of the test results showed that the film breakage rate $y_1$ and the knotted feature removal rate $y_2$ were 88.33% and 87.67%, respectively. It showed that the verification test results are basically consistent with the model prediction results.

### 4. Discussion

(1) The test shows that the cutting-support motor speed ratio and the sliding cutter edge angle have a significant impact on the film breakage rate. After analysis, the following can be seen: when the cutting-support motor speed ratio is too small, the phenomenon of missing cutting is serious, so the film breakage rate is too low; when the cutting-support motor speed ratio is too large, the material on the sliding cutter will slide and gather without enough time, resulting in a poor crushing effect; when the sliding cutter edge angle is too large, the force component of the material in the normal direction is too small, and the crushing effect will be poor; when the sliding

cutter edge angle is too small, the material will not slip, so the crushing effect will be poor.

(2) The test shows that the support motor rotation speed and the sliding cutter edge angle have a significant influence on the removal rate of the knotted feature. When the support motor rotation speed is fast, the rotation speed of the knotted feature on the support rod is too fast, resulting in an insufficient removal time during the mixing process. When the speed of the support motor is too slow, the material slips on the support rod due to gravity, and it is too late to be crushed. When the angle is too large, the component force of the material in the normal direction is too small, resulting in a poor removal effect of the knotted feature.

(3) The experiment only studies the crushing effects of the film, such as film breakage rate and knotted feature removal rate in the waste, but regarding the entanglement of the waste during the crushing process and the broken film in the air blowing flow. The characteristics of the migration motion under the synergistic effect of the field and the rotational motion still need to be further studied.

(4) For the knotted features of the plastic mulch waste, an inner arc cutter was proposed. Compared with the V-shaped cutter designed in reference [18], the inner arc cutter is more effective in breaking the knotted features of the waste. However, the cutter in reference [18] is more effective for the overall cutting uniformity of the waste. Compared with the water-washing separation in reference [19], the sliding cutting device is more suitable for residual film treatment companies, and the method used in reference [19] has a higher waste of water but a better separation effect.

## 5. Conclusions

Aiming at the technical bottleneck of the difficulty in crushing the knotted feature of the membrane impurities, an internal blowing and sliding cutting crushing device was designed. Based on the analysis of the combination of theoretical simulation, the different installation methods were theoretically studied, and the key parameters affecting the crushing effect were obtained. The center combination method was used to carry out the crushing test of the mechanically harvested membrane impurity waste, and the film breakage rate and the knot feature removal rate of the membrane impurity waste after crushing were calculated. Regression analysis of variance was used to construct the relationship model between the key parameters of the crushing device and the film breakage rate and the removal rate of the knotted features. The model coefficient $p$ values of the constructed second-order response models were all <0.01, the coefficients of determination $R^2$ were all >0.95, and the coefficients of variation C.V. were all >1.38%, indicating that the constructed models were significant and well explained, and could accurately and reliably predict the test response indicators. At the same time, through the optimization analysis of the quadratic regression model, the best parameter combination obtained was as follows: support motor speed $n_1$, 120 r·min$^{-1}$; cutting-support motor rotation speed ratio $\lambda$, 5.3; cutting frame-support frame rotation cutter edge angle, 52°. Under the same conditions, the errors between the physical test value and the model prediction value were 2.17% and 3.52%, respectively, indicating that the verification test results were basically consistent with the model prediction results.

**Author Contributions:** Conceptualization, M.G. and H.B; methodology, M.G.; software, L.X.; validation, M.G. and Y.C.; formal analysis, M.G.; investigation, X.L.; resources, M.G.; data curation, C.Y.; writing—original draft preparation, M.G.; writing—review and editing, B.H.; visualization, Y.C.; supervision, B.H.; project administration, B.H.; funding acquisition, X.L. All authors have read and agreed to the published version of the manuscript.

**Funding:** This research was funded by the National Natural Science Foundation of China Regional Project Fund (51865051 and 52265037), the Open Project of the Key Laboratory of the Modern Agricultural Machinery Corps (BTNJ2019003), and the Youth Innovation Cultivation Program of the School of Mechanical and Electrical Engineering of Shihezi University (KX01230202).

**Institutional Review Board Statement:** Not applicable.

**Informed Consent Statement:** Not applicable.

**Data Availability Statement:** The data presented in this study are available on demand from the first author at (shzdxjdxyg-my@shzu.deu.cn).

**Conflicts of Interest:** The authors declare no conflict of interest.

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
