# Peer review of "Design and Test of a Sliding Cutting Device for the Plastic Mulch Waste"

_sustainability, doi:10.3390/su15054513_

Round 1
Reviewer 1 Report
The article entitled “Design and Test of a Sliding Cutting Device for the Plastic Mulch Waste” is much-needed research for sustainable Mulch farming.
Abstract:
Line 16: “. Conducted a three-factor five-level orthogonal test and performed 16 regression variance analysis”. Please specify the design type with software name.
Introduction:
Please mention if such type of product previously designed or not, if yes then how this one is different.
Materials and methods:
The cutter was made of which material- Iron, steel or any other? What the effects of material on cutting capacity?
Table 1, line 250: Please mention variables a, b, c and responses y1 and y2 in table itself at the end as footnote. Moreover, what the values in brackets at some places in column a, b, c indicates?
Line 267: Please mention Design expert version.
r·min-1 is wrongly typed as r·min-1 at several places in whole MS.
Results:
In Table 2 what the value 5.907E-3 represents? Please see it. Also mention C.V. full form in footnote.
Figure 10, x-axis values not clearly visible. Please provide improved, high-resolution image of it.
Discussion:
There are no references in results and discussion section. Is there no such or related previous study!!! Without references how to compare the results.
Reviewer 2 Report
The study was carried out to solve the problem of mechanically recycled plastic agricultural mulch waste which is caused by the crop straw residues knotting in the plastic film, which decreases the efficiency of cutting. The authors designed the special cutting device for the mixed plastic and straw wastes and computed the best parameters, including supporting motor speed of the device, cutting supporting rotation speed ratio, cutting edge angle, impacted the film breakage rate and knotted feature removal rate. The best parameter combination was chosen due to the model, and the verification test was carried out to prove the reliability of the model. The errors between the physical test value and the model prediction value of the two response indicators were 2.17% and 3.52%, respectively.
In general, the given study suits one of the major areas of interest of the Waste and Recycling Section of the Journal, namely, Treatments aiming at the recovery of materials from waste.
The manuscript is designed very well, the methodological part adequately describes all the computations, and the results are clearly explained. The authors did not compare their results to other investigations in the field because, as far as I have understood, they proposed such a device for the first time. However, I have one question to be answered in the introduction. It was acknowledged from the literature that the problem of recycling a big amount of plastic film after mulching, is serious. Also, that the mechanized collection is the main technology used to remove the film. I also found some additional articles in this area. But after this introduction the authors claim the problem of knots between the film and the straw in lines 36-39. And there is no single reference to confirm this problem. No data, no statistics, no survey, no anything. I quickly googled “plastic mulch knots straw” and found nothing. The first step in every project is to find who and how solves this problem for today. I am pretty sure that the problem exists, but the authors need to prove it and to show the scale.
